# Zero-Shot 3D Drug Design by Sketching and Generating

**Siyu Long**[*1]**, Yi Zhou**[2]**, Xinyu Dai**[1]**, Hao Zhou**[3]

[1]National Key Laboratory for Novel Software Technology, Nanjing University
[2]ByteDance AI Lab
[3]Institute for AI Industry Research (AIR), Tsinghua University
`longsy@smail.nju.edu.cn`, `zhouyi.naive@bytedance.com`
`daixinyu@nju.edu.cn`, `zhouhao@air.tsinghua.edu.cn`

## Abstract

Drug design is a crucial step in the drug discovery cycle. Recently, various deep learning-based methods design drugs by generating novel molecules from scratch, avoiding traversing large-scale drug libraries. However, they depend on scarce experimental data or time-consuming docking simulation, leading to overfitting issues with limited training data and slow generation speed. In this study, we propose the zero-shot drug design method DESERT (**D**rug d**E**sign by **Sk**Etching and gene**R**a**T**ing). Specifically, DESERT splits the design process into two stages: sketching and generating, and bridges them with the molecular shape. The two-stage fashion enables our method to utilize the large-scale molecular database to reduce the need for experimental data and docking simulation. Experiments show that DESERT achieves a new state-of-the-art at a fast speed.[1]

## 1 Introduction

Drug design is a crucial step in the drug discovery cycle, which is the inventive process of finding new drugs based on a biological target (usually a protein pocket) [1, 2, 3]. However, seeking appropriate drugs for a particular target is quite challenging due to the enormous space of drug candidates (almost $10^{33}$). Traditional drug design approaches usually employ virtual screening [4, 5, 6] and molecular dynamics [7, 8] to traverse in a large scaled drug library, which is time-consuming and could not produce novel drug candidates. Recently, a line of work proposes to realize drug design by generating drug molecules from scratch using deep generative models [9, 2, 10], which is quite promising due to the fast speed and the ability of *de-novo* drug design.

Most of current drug generation model are developed upon 1D (SMILES) [11, 12, 13] or 2D (molecular graph) [14, 15, 16, 17, 18, 19] molecular structures, which heavily rely on expensive experimental data for supervised training while ignoring the 3D interaction information between the drug and the pocket. In a word, they attempt to find a molecule that maximizes the score given by a bio-activity predictor trained on experimental data. They employ different optimization methods, such as Generative Adversarial Network (GAN) [20], Bayesian Optimization (BO) [21, 22, 9], Reinforcement Learning (RL) [23, 24, 20, 25, 26, 27], Evolutionary and Genetic Algorithms (EA/GA) [28, 29, 30, 31, 32], and Markov Chain Monte Carlo (MCMC) [33, 34], in the molecular space for obtaining desired drug molecule under certain constraints. However, we argue that a purely data-driven approach of drug design is practically limited, since in most cases, the quantity of experimental drug-pocket pairs hardly enables supervised training of *de-novo* drug design. Generally, most protein pockets lack bio-activity data, and learning on noisy and deficient data may lead to severe overfitting problems.

---

[1]Code is available at https://github.com/longlongman/DESERT.

∗ Work was done when Siyu Long was a research intern at Bytedance AI Lab.

36th Conference on Neural Information Processing Systems (NeurIPS 2022).

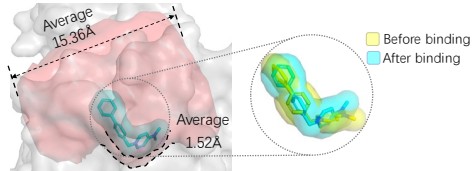

Figure 1: When a drug is binding to a pocket, its shape does not change too much (with an RMSD less than 1.391Å) and is complementary to the pocket.

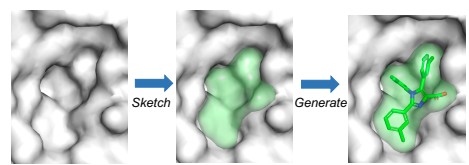

Figure 2: DESERT splits the whole drug design process into two stages: sketching the shape and generating the molecules.

Recently, several drug design models have been proposed to directly generate drug molecules in the 3D space, the realistic space of drug-target interaction. Generating in such space is very promising for the potential of leveraging some prior knowledge (e.g., physical knowledge) instead of entirely counting on data-driven methodology. Specifically, Masuda et al. [35] and Luo et al. [2] propose efficient 3D generative models to learn the atom density conditioned on protein pockets, with GAN and auto-regressive models, respectively. Nevertheless experimental data are still obligatory in their models for achieving satisfactory results. Even more noteworthy is GEKO [10], which combines the ideas of 3D generation and physical simulation to obtain state-of-the-art drug design performance without the help of large-scale experimental data. Intuitively, GEKO performs geometric editing in the 3D molecule space guided by docking simulation (physical knowledge) [36, 37]. However, GEKO may suffer from two concerns: a) the frequent invocation of docking is very time consuming, which significantly slows down the speed of the drug design model [38, 39]. b) docking may not always be accurate enough, especially in some complex settings [40, 41]. In such a case, being heavily dependent on the docking accuracy could hurt the generalization of the proposed drug design model.

In this paper, we propose the zero-shot approach DESERT, namely **D**rug d**E**sign by **Sk**E**tching and gene**R**a**T**ing. Motivated by the idea of *structure determines properties* [42, 43, 44, 45], DESERT is built on the assumption that molecular shape determines bio-activity between drug molecules and its target pocket. In other words, we suppose that a drug candidate would have satisfactory bio-activity to a target pocket if their shapes are complementary (see Figure 1).

With such prior, as shown in Figure 2, DESERT splits the whole drug design process into two stage: sketching and generating, which employs the *molecular shape* as the bridge of the two stages. Such splitting makes DESERT enjoy two advantages: a) DESERT does not heavily rely on docking simulation, which only optionally uses docking for post-process and thus avoids the aforementioned disadvantages of GEKO. b) DESERT abandons the expensive experimental data. Specifically, in the sketching stage, we only need to sample some reasonable shapes complementary to the target pocket. In the generating stage, DESERT proposes to employ a generative pre-trained model (from shape to concrete molecular) to fill the shape obtained in the last sketching stage. Notably, the generative pre-trained model is only trained on the ZINC database, which contains 1000M pairs of molecules and their corresponding shapes. This process does not rely on experimental data, making DESERT work in a zero-shot fashion. [2]

Note that DESERT is not baseless. Besides the idea that structure determines properties (an important concept of Structural Biochemistry), we also have preliminary results to verify our assumption. Figure 1 shows that after binding to a pocket, the root-mean-square deviation (RMSD) is not too shabby (1.391Å) compared with molecular conformation generation methods such as CGCF [46] (1.248Å) and ETKDG [47] (1.042Å). The results suggest that the molecular shape is stable when molecules bind to proteins. We also show in Figure 1 that a ligand often attaches tight to a pocket, which means their shape is complementary.

In summary, our contributions are three-fold: (1) We propose DESERT, a generative method for *de-novo* 3D shape-based drug design in the zero-shot setting. (2) DESERT trains a pre-trained model

---

[2]The generation stage of DESERT can also be equipped with chemistry priors, we conduct some experiments in Appendix 2.2.

**Training**
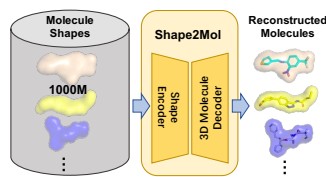

**Designing (Ligand-based)**
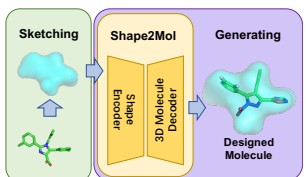

**Designing (Pocket-based)**
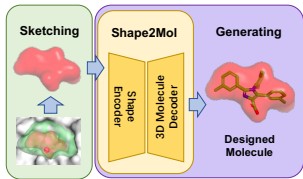

(a) Pre-training with massive unbound molecules

(b) DESERT with ligand-based sketching

(c) DESERT with pocket-based sketching

Figure 3: Overview of DESERT. In Figure (a), we use massive unbound molecules[2] to train the SHAPE2MOL, which includes two sketching variants. In Figure (b) and (c), we use the SHAPE2MOL to generate 3D molecules to fill in the given shapes. In Figure (b) where existing drugs are available, we treat their shapes as desired molecule shapes (ligand-based). In Figure (c) where only protein pockets are given, we heuristically sample reasonable molecular shapes from the pockets (pocket-based).

from massive unbound molecules [3], eliminating the constraints of labeled data. (3) DESERT achieves a new state-of-the-art result (the docking score improves $0.79\mathrm{kcal/mol}$ and $2.93\mathrm{kcal/mol}$ over the best supervised method on two datasets) at a fast speed (about 20 times faster than GEKO[4]).

## 2 Proposed Method: DESERT

In this section, we describe the proposed method in detail. Inspired by the two aforementioned preliminary studies (see Figure 1 and Figure 2), DESERT designs drugs for proteins in a two-stage fashion and employs the shape as a bridge since previous work has shown the feasibility of designing drug by molecular shape [49, 50, 51, 52, 53].

Specifically, DESERT designs drugs by first sampling appropriate shapes complementary to the target pocket and then mapping the shapes to specific molecules. In Section 2.1, we first introduce the overall picture of how DESERT works in a zero-shot setting. Then we pose two challenges: how to sketch the reasonable molecular shapes and how to generate corresponding molecules based on the shapes. We put forward solutions in Section 2.2 and Section 2.3, respectively.

### 2.1 Zero-Shot Pipeline

In this section, we focus on how DESERT designs drugs in the zero-shot setting. Briefly speaking, DESERT produces molecules in a two-stage fashion: sampling the shape of the desired drug first (sketching) and generating molecules conditioned on the resulting shape (generating).

**Zero-Shot Sketching** There are mainly two cases when DESERT needs to sample molecular shapes. In the zero-shot case where no reference protein ligands are available, DESERT samples reasonable shapes from protein pockets (see Figure 3c) based on biological observations. Besides, DESERT can also reuse the shape of a ligand to design a novel one (see Figure 3b). Details are listed in Section 2.2.

**Zero-Shot Generating** DESERT generates molecules through a pre-trained generative model, namely SHAPE2MOL, which can convert a given shape into diverse molecules (see Figure 3a). In this procedure, we utilize massive unbound molecules to train the model. Thus no information about proteins is needed. Details of this model are presented in Section 2.3.

### 2.2 Sketching Molecular Shapes

The sketching stage is responsible for deciding what desired molecules look like. In this section, we show how we design two heuristic methods to sketch the shapes of desired molecules.

---

[3]For clarity, the following terms will be used throughout this study: "bound drug/molecule" (or "unbound drug/molecule") refers to the drug/molecule that is bound (or unbound) to proteins [48].

[4]We calculate the speed by measuring the time from given a pocket to getting 100 molecules.

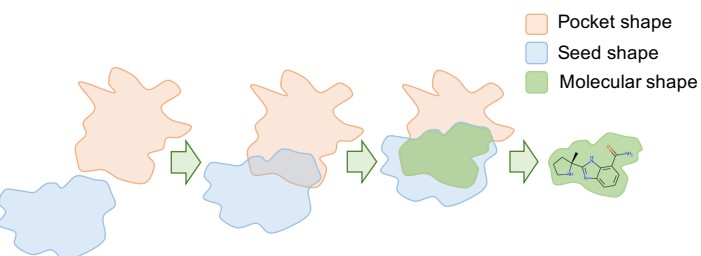

Figure 4: A 2D illustration of sampling molecular shapes from pockets. The 2D molecule in the figure represents a potential drug that fits the molecular shape.

There are mainly two cases when we sketch a molecule shape based on whether the ligand is provided. When the ligand is available, the sketching process can be trivial since molecules with similar shapes have similar properties. It is reasonable to directly use the ligand's shape as the shape of the desired molecule, which we call **Ligand-based Sketching**. If the ligand of the pocket is unavailable, the challenge is how to obtain a shape that has a high potential to bind to a pocket. [5] We name this **Pocket-based Sketching**, and our main idea is to sample a region with an appropriate size complementary to the surface of the pocket.

Our idea for Pocket-based Sketching is based on two main observations:

1. Ligands mainly lie in the area close to the pocket surface. Figure 1 shows that the shape of satisfactory ligand is tightly complementary to the pocket.
2. Pockets are usually much larger than ligands, suggesting that directly utilizing the shape of pockets to design molecules is inappropriate.

To this end, we present an algorithm (see the Appendix 1.1) to obtain the desired molecule shape, which is of the appropriate size and complementary to the pocket surface. We achieve this goal by finding another shape (namely seed shape) that intersects with the pocket, where the intersection has a similar size to a molecule. We also show a 2D illustration in Figure 4.

## 2.3 Generating 3D Molecules by SHAPE2MOL

In this section, we introduce SHAPE2MOL, an encoder-decoder network mapping a shape to diverse and high-quality 3D molecules. There are plenty of unbound molecule data, e.g., 1000M molecules in the ZINC database, making it possible to learn a large-scale pre-trained generative model from shape to molecule.

Concretely, we formulate the problem as an image-to-sequence generation, where the shape is voxelized as a 3D image (see 2.3.1), and the 3D molecule is converted to be a sequence (see 2.3.2). Our generative approach is capable of modeling any complicated molecule structure and the linearization makes large-scale pre-training easier to implement.

### 2.3.1 Encoder: Voxelized Shape

The shape encoder is a 3D extension of the ViT [56], where we use 3D patches instead of 2D patches in the original ViT. Let $\mathcal{A}$ denote the set of all atoms, a molecule $m$ can be constructed as a collection of atoms and their corresponding coordinates:

$$m = \{(a, c) | a \in \mathcal{A}, c \in \mathbb{R}^3\}$$

Given a molecule $m$, we transform its shape into a 3D image with a voxelization function $v_m : \mathbb{Z}^3 \to \{0, 1\}$:

$$v_m(x, y, z) = \begin{cases} 1 & \exists (a, c) \in m, \|(x, y, z) - c\|_2 \leq r(a) + \epsilon \\ 0 & \text{otherwise} \end{cases}$$

where $r$ denotes the Van der Waals radii [57], $\epsilon$ is a perturbed noise which helps prevent overfitting [58].

---

[5]The pocket can be generated by CAVITY [54] or f-pocket [55]. In this study, all protein pockets are generated by CAVITY.

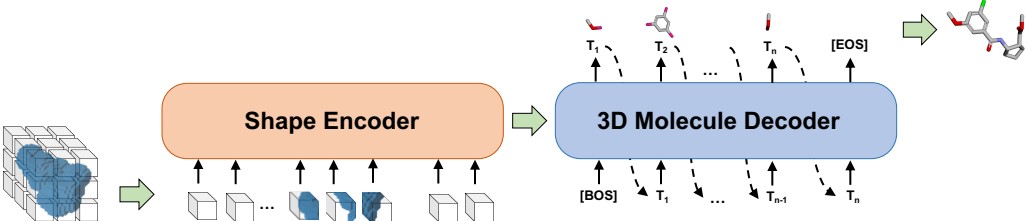

Figure 5: The architecture of SHAPE2MOL.

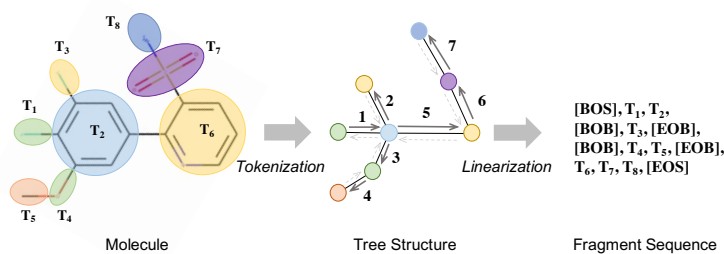

Figure 6: The process of converting a molecule into a sequence.

### 2.3.2 Decoder: Linearized Molecule

The molecule decoder in our model is similar to the Transformer decoder in machine translation [59]. The main difference is that the decoding object here is a 3D molecule instead of a 1D sequence. To address this, we propose a 3D molecule decoder, which handles a 3D molecule as a sequence of tuples. The sequence object eases the implementation of a pre-trained model. To obtain the object, we first cut a molecule into pieces, then convert it to a sequence.

**Tokenization** We cut a molecule into pieces so that the generative process can be easily factorized. Our principles are three folds: (i) preserving the functional groups since they are vital for determining molecule properties [60], (ii) avoiding too large size of the vocabulary to ease the pre-training process [61], (iii) no circles exist in the segmented molecules since a tree structure is simpler to handle than a graph. Our method is simple yet efficient: first tokenizing the molecules with BRCIS [62] and then cutting all single bonds attached to a ring. More details about some pilot experiments can be found in Appendix 1.2.

**Linearization** For the network output, we propose to utilize a linearized sequence to represent the target molecule graph, which is not only convenient for training, but also has the strong power to represent any complicated tree structure. We first select an fragment whose degree is 1 as the root of the tree (e.g., T1, T3, T5, and T8 in Figure 6). Then we traverse the tree in the depth-first-traverse style[63]. Whenever we enter or leave a branch, we will add two special symbols [BOB] and [EOB] (beginning/ending of a branch), respectively.

More particularly, we use a tuple $(C, P, R)$ to represent a fragment $F$, where $C = \mathbf{1}_F$ is an indicator function [64] denoting its index in the vocabulary, $P \in \mathbb{R}^3$ is the translation vector [65], $R \in \mathbb{R}^4$ is the rotation quaternion [66]. In order to stabilize the training process, we further discretize the continuous variable $P$ and $R$ into $P^c$ and $R^c$, respectively. Taking the translation vector $P$ as an example, we convert it into a binary vector $P^c$, which satisfies:

$$P^c[i] = \begin{cases} 1 & \lfloor \frac{L}{b}i \rfloor \leq P < \lceil \frac{L}{b}i \rceil \\ 0 & \text{otherwise} \end{cases}$$

where $L$ is the max translation length, $b$ is the bin size.

## 2.4 Training & Decoding

**Training** Given the output probabilities of the model $(\hat{C}_i, \hat{P}_i^c, \hat{R}_i^c)$, which denote the probability of a fragment, a discreterized translation vector, and a discreterized rotation vector, respectively. We calculate the corresponding cross-entropy loss and use their sum as the final loss function.

$$\mathcal{L} = -\sum_{i=1}^{n} \left\{ C_i \log \hat{C}_i + P_i^c \log \hat{P}_i^c + R_i^c \log \hat{R}_i^c \right\} \tag{1}$$

where $n$ is the length of fragment sequence.

We use 100M unbound molecules sampled from the lead-like subset of ZINC as the training data. The Transformer's dimension is 1024, and both encoder and decoder have 12 stacked Transformer layers. When training SHAPE2MOL, we set the dropout rate as 0.1, batch size 2048, train step 300K and use AdamW [67] with learning rate 5e-4, weight decay 1e-2, and warmup step 4000 as the optimizer. The model is developed by ParaGen [68] [6] and trained on 32 Telsa V100 GPU cards for 2 weeks. Following [35], we also randomly rotate and translate the input shape for invariance.

**Decoding** We design a decoding strategy to provide diverse and high-quality candidate molecules for a given protein. To achieve diversity, we employ the sampling method Nucleus [69] to generate multiple fragment sequences. Then we convert the sequences back to molecules with a greedy algorithm (see Appendix 1.2), which connects the fragments by greedily enumerating the nearest pair of breakpoints. [7] Finally, we do some post-processing operation to further improve the diversity and quality. We remove the duplicate molecules and leverage the docking simulation to drop molecules that do not pass the affinity threshold.

When testing SHAPE2MOL, we set the threshold of Nucleus sampling to 0.95. For each protein pocket, we sketch 200 shapes. For each shape, we generate 1000 molecules. More details of SHAPE2MOL can be found in the Appendix 1.3.

## 3 Results and Discussions

### 3.1 Experiments

**Data** We evaluate the performance of our method on drug design by using a total of 12 proteins (PDB IDs: 1FKG, 2RD6, 3H7W, 3VRJ, 4CG9, 4OQ3, 4PS7, 5E19, 5MKU, 3FI2, 4J71), which is a combination of the test data used in Masuda et al. [35] and Jin et al. [23]. Among the proteins, only JNK3 and GSK3$\beta$ (PDB IDs: 3FI2, 4J71) have sufficient labeled data to train well-performing bioactivity predictors. We denote the set of these two proteins as Set B and the rest as Set A. Because 1D/2D methods need bioactivity predictors to design drugs for given proteins, we only evaluate them on Set B while evaluating 3D-based methods on both sets.

**Baselines** We compare DESERT with some baselines for drug design. Based on the resource needed for designing drugs, we further divide these baselines into three groups: **Guided** methods need docking simulation or extra bioactivity predictors to provide supervision signals. **Supervised** methods rely on labeled data to train their models. **Retrieved** methods directly search the database for desired molecules.

**Evaluation** Following Yang et al. [10], we evaluate the performance of methods from two aspects. (1) the molecular space covered by designed results. (2) the capacity to provide highly active molecules. As a high-quality molecular space should contain diverse, novel molecules with pharmaceutical potential, we use five metrics to evaluate the molecular space: Uniqueness (**Uniq**), Novelty (**Nov**), Diversity (**Div**), Success rate (**Succ**), and Product (**Prod**). To evaluate the ability to provide highly active molecules, we compare the distributions of $\text{Vina}_{\text{score}}$ and use Median Vina Score (**Median**) to quantify the distribution. More details about these metrics can be found in Appendix 2.1.

**Detail of Generation** We describe how different methods generate molecules for comparison. For each protein, every method needs to generate 100 molecules for comparison. For 3D methods, we

---

[6]https://github.com/bytedance/ParaGen

[7]Whenever we cut a chemical bond in tokenizing, we mark the atoms of the chemical bond as breakpoints.

Table 1: Performance comparison among drug design methods. ↑ indicates higher is better. ↓ indicates lower is better.

| Targets | Method | | Uniq (%)↑ | Succ (%)↑ | Nov (%)↑ | Div ↑ | Prod ↑ | Median (kcal/mol)↓ |
|---|---|---|---|---|---|---|---|---|
| Set A | Guided | GEKO [10] | 100.0 | 55.7 | 100.0 | 0.912 | 0.51 | -9.58 |
| | Supervised | liGAN [35] | 100.0 | 0.4 | 100.0 | 0.924 | 0.00 | -5.84 |
| | | 3D SBDD [2] | 69.7 | 13.6 | 98.9 | 0.839 | 0.08 | -8.83 |
| | Retrieved | SCREEN (1K) | 100.0 | 25.6 | 100.0 | 0.892 | 0.23 | -7.46 |
| | | SCREEN (200K) | 100.0 | 64.0 | 100.0 | 0.889 | **0.57** | -8.66 |
| | Ours | DESERT-LIGAND | 100.0 | 65.3 | 87.0 | 0.786 | 0.41 | -8.89 |
| | | DESERT-POCKET | 100.0 | 61.1 | 100.0 | 0.908 | **0.57** | **-9.62** |
| Set B | Guided | JT-VAE [9] | 100.0 | 13.0 | 100.0 | 0.907 | 0.12 | -8.35 |
| | | RationaleRL [23] | 100.0 | 27.0 | 35.0 | 0.884 | 0.08 | -7.75 |
| | | GA + D [30] | 39.0 | 24.0 | 87.0 | 0.852 | 0.06 | -7.22 |
| | | GraphAF [26] | 97.0 | 0.5 | 100.0 | 0.946 | 0.00 | -4.22 |
| | | MolDQN [27] | 76.5 | 0.0 | 100.0 | 0.742 | 0.00 | -5.52 |
| | | MolEvol [70] | 99.5 | 40.5 | 63.5 | 0.742 | 0.17 | -8.19 |
| | | MARS [33] | 86.0 | 31.5 | 93.0 | 0.805 | 0.22 | -7.68 |
| | | GEKO [10] | 100.0 | 57.0 | 100.0 | 0.910 | 0.52 | -9.19 |
| | Supervised | liGAN [35] | 99.8 | 0.2 | 100.0 | 0.923 | 0.00 | -5.34 |
| | | 3D SBDD [2] | 99.9 | 5.2 | 100.0 | 0.853 | 0.05 | -6.39 |
| | Retrieved | SCREEN (1K) | 100.0 | 3.0 | 100.0 | 0.891 | 0.03 | -6.94 |
| | | SCREEN (200K) | 100.0 | 32.0 | 100.0 | 0.882 | 0.28 | -7.95 |
| | Ours | DESERT-LIGAND | 100.0 | 18.0 | 100.0 | 0.913 | 0.17 | -7.34 |
| | | DESERT-POCKET | 100.0 | 61.0 | 100.0 | 0.907 | **0.55** | **-9.32** |

further use the local minimization module in Vina [36] to optimize the generated structures. For 1D/2D methods, we first use RDKit to generate the 3D conformer of their results. For method SCREEN, we sample 1K and 200K molecules from ZINC and return 100 molecules with the highest Vina$_{score}$ as the generated results.

## 3.2 Main Results

Table 1 shows the results of DESERT and baselines. All values are averaged over target proteins. Our main findings are listed as follows:

*I. The zero-shot* DESERT *achieves the SOTA result at a fast speed.* DESERT's performance is strong compared with GEKO, an MCMC-based model with a huge sampling space. Compared with GEKO, which works in a trial-and-error way, DESERT makes a more clever choice by pruning the space with its biological knowledge regarding the shape and quickly finds a good solution with limited hints from a teacher. We compare the generation speed in Figure 7.

*II. The shape helps* DESERT *produce high quality molecules.* The molecular space of SCREEN is the ZINC database, while that of DESERT is generated by the pre-trained model SHAPE2MOL, which is aware of the pocket's shape. According to the table, both DESERT-LIGAND and DESERT-POCKET show observably better performance than their counterparts, i.e., SCREEN (1K) and SCREEN (200K).

*III. More comprehensive exploration of protein pockets benefits performance.* Instead of using the molecular shapes of reference ligands as input, DESERT-POCKET comprehensively explores the protein pockets by sampling multiple molecular shapes from them. Therefore, it has the potential to obtain diverse, high-quality molecules that bind to protein pockets in different regions. While DESERT-LIGAND only considers one region, i.e., the region that ligands lie in, limiting the exploration of protein pockets. We show a case in Figure 8.

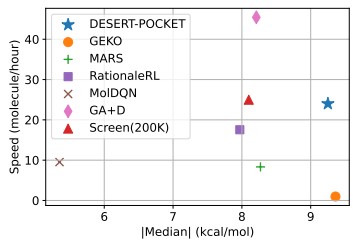

Figure 7: The absolute value of the median vina score and the speed of different methods on target protein 3FI2.

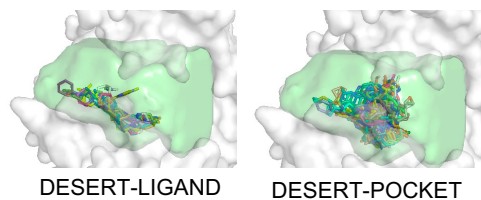

Figure 8: Molecules generated by DESERT on protein 4CG9.

Table 2: Comparison of shape faithfulness and structure rationality. "Random" is the Shape Tanimoto between two random molecules, "Real" is the Free Energy of reference ligands.

| Method | Shape Tanimoto | Free Energy (kcal/mol) |
|---|---|---|
| Random | 0.325 | / |
| Real | / | 167.28 |
| liGAN (Ligand) | 0.869 | 289.55 |
| DESERT-LIGAND | 0.875 | 188.54 |

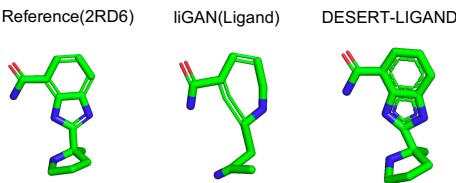

Figure 9: A cases from liGAN and DESERT.

*IV. Unsupervised methods have larger potential than the supervised counterparts.* As labeled data is inadequate, e.g., scPDB [8] only has 16,034 entries, the supervised methods easily collapse to the main molecule pattern in their dataset. When generating molecules with them, we often get the same molecules, e.g., 3D SBDD can only generate 16 unique molecules for protein 4OQ3. DESERT utilizes massive unbound molecules, which leads to the learned space being denser. Combing with an appropriate sampling method, it can generate diverse molecules.

## 3.3 Comparison with Related Shape-based Models

In this section, we study the shape faithfulness and structure rationality compared with previous shape-based models. In Table 2, we use Shape Tanimoto [9] to evaluate the faithfulness and use Free Energy to quantify the rationality [74]. We compares DESERT-LIGAND with liGAN (Ligand), a variant of liGAN [35] which utilizes the existing ligands. Although liGAN (Ligand) achieves high performance on Shape Tanimoto, its atom-based decoding strategy does not guarantee the correct relative position between atoms. Therefore, liGAN shows higher Free Energy, indicating that unrealistic structures may appear. Figure 9 shows a case where liGAN produces an ill ring structure.

## 3.4 Ablation Study of Generating

In this section, we evaluate several designs of the generating stage in our DESERT method, which relate to the pre-trained model and decoding strategy. All results are based on an extra test set from ZINC and a smaller version of SHAPE2MOL. [10]

**Pre-training Configuration** We evaluate the model quality on different pre-training configurations (mainly focusing on the size of the model and training data). The results in Figure 10 show: (1) Larger model achieves better performance. Devlin et al. [75] observes similar phenomenons in natural language modeling. (2) Performance saturation occurs when the dataset is of moderate size. As the

---

[8]scPDB is a high-quality labeled dataset for 3D drug design.

[9]Shape Tanimoto [71, 72, 73] measures the similarity between the input and generated molecules. Shape Tanimoto$(A, B) = \frac{A \cap B}{A \cup B}$, where $A$ and $B$ are two molecular shapes.

[10]The extra test set contains 10K molecules. The smaller SHAPE2MOL has 512 model dimension and 6 layers of encoder and decoder. We use a greedy decoding strategy and remove post-processing.

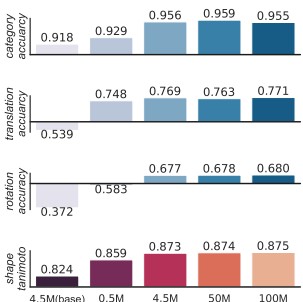
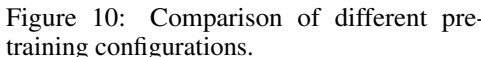
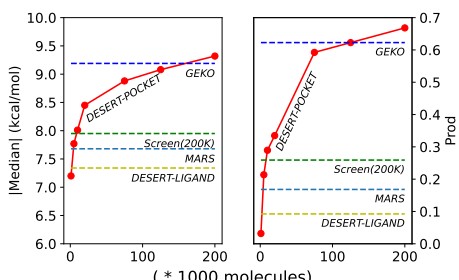

Figure 10: Comparison of different pre-training configurations.

Figure 11: Comparison of sampling space size.

Table 3: Performance comparison on more protein targets. Because the original test data only contains incomplete protein pockets, we recover the complete pockets by aligning the incomplete pockets to the structures from PDB database. We managed to recover 55 pockets and apply DESERT-POCKET to them. For 3D SBDD, we use the released code and apply the same post-processing used in DESERT to it.

| Metric Med. | 3D SBDD w/o post-processing | 3D SBDD w post-processing | DESERT-POCKET w/o post-processing | DESERT-POCKET w post-processing |
|---|---|---|---|---|
| Vina Score (kcal/mol) | -6.069 | -7.584 | -6.148 | -9.410 |
| QED | 0.522 | 0.501 | 0.614 | 0.549 |
| SA | 0.672 | 0.623 | 0.612 | 0.616 |
| Diversity | 0.873 | 0.826 | 0.926 | 0.908 |

map from radii to atoms is easy to learn, the model can capture it with a moderate dataset. Liu et al. [76] reports a similar result that a large dataset does not necessarily lead to better quality.

**More Ablation** We also do ablation study about some model variants (including *discretization* and *robust training*) and *decoding strategies*. We further study *chemical information driven design* and *atom-based pre-training*. We refer the reader for more details to the Appendix 2.2.

### 3.5 Ablation Study of Sketching

In this section, we study the sketching stage in our DESERT method, which includes the effect of sampling space size and seed shape on the method's performance. All the results are based on the same DESERT-POCKET method in Section 3.2 and are calculated on Set B.

**Sampling Space Size** In Figure 11, we evaluate the performance of DESERT with respect to sampling space size, i.e., the total number of generated molecules before post-processing. The results show: (1) Increasing sampling space size leads to better performance. With a larger sampling space, DESERT finds more molecular shapes complementary to pockets, leading to a performance rise. (2) The shape can effectively prune the sampling space for screening. Instead of directly searching molecular space, DESERT achieves a similar performance by pruning the space from 200K to 10K with molecular shapes.

**More Ablation** We also do ablation study about the usage of different *seed shape*. We refer the reader for more details to the Appendix 2.3.

### 3.6 Apply DESERT to More Protein Targets

To test the generalization ability of our method more widely, we also apply DESERT to the test data from Luo et al. [2], which contains 100 protein targets. As shown in Table 3, based on the idea of

*structure determines properties*, DESERT-POCKET generalizes well on different target proteins in both setting, i.e., with/without post-processing. Supervised methods, like 3D SBDD, hindered by scarce training data, can not generate diverse molecules. In contrast, training on massive unbound and drug-like molecules, DESERT easily generates diverse and promising molecules. Moreover, sketching molecular shapes based on given pockets also contributes to the better binding affinity of DESERT. However, DESERT gives a lower SA score than 3D SBDD. We assume that it is because the generated molecules of DESERT tend to be structurally complicated, which leads to a slightly worse synthesis score.

## 4 Conclusions

In this study, we propose a zero-shot drug design method DESERT, which splits the drug design process into two stages: sketching and generating. DESERT bridges the two stages with the molecular shape and utilize a large-scale molecular database to reduce the dependence on experimental data and docking simulation. Experiments show that DESERT achieves a new state-of-the-art at a fast speed.

## Acknowledgements

We would like to thank the anonymous reviewers for their insightful comments. Both Hao Zhou and Xinyu Dai are the corresponding authors. This work is jointly supported by Guoqiang Research Institute General Project, Tsinghua University (No. 2021GQG1012) and National Science Foundation of China (No. 61936012 and 61976114).

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
