# OpenReview forum: "Zero-Shot 3D Drug Design by Sketching and Generating"
_NeurIPS.cc/2022/Conference — NeurIPS 2022 Accept_

### Official Review · Reviewer_Qdyd · 2022-06-28

**Rating:** 7
**Confidence:** 5
**Soundness:** 3 good
**Presentation:** 3 good
**Contribution:** 3 good

**Summary:**

This paper presents a new framework and pre-training scheme for designing pocket-conditioned ligands. Specifically, it leverages the shape (voxel) of ligands and protein pockets in pre-training a sketching network that specifies the shape of the protein pocket, and learning a shape2Mol network that predicts specific ligands that fit into the given voxel shapes. The model were trained over billions of molecules in ZINC database and achieved state-of-the-art results.

**Questions:**

See above, I put questions related to each point in the strengths and weaknesses part.

**Limitations:**

The limitations are not discussed in the paper but may come from two sides: (1) whether this method can generate diverse ligands that bind to protein targets, (2) the efficiency of the method. The first one concerns whether the formulation makes sense in the biological context, or if not, is still a good way to leverage unlabeled data. The second one needs a bit more experiments to support.

**Strengths And Weaknesses:**

Strength
* This paper proposes an interesting idea to recognize the voxel/shape of the protein pocket and leverage the information for pre-training. It is quite different from the other line of work which leverages the atoms of both proteins and ligands that aim to learn the interaction for generative design. It is interesting to see how this idea plays out and I would suggest adding more discussions about how it is considered in traditional structure-based drug design, shape/voxel vs. atomic interaction.
* The proposed idea is proper in real-world scenario, where the paired ligand and pocket data are limited, thus I am convinced pre-training is needed.

Weakness
* How is sampling achieved for generating diverse molecules for a specific pocket? In the paper, only part mentioning it refers to Nucleus [68], does it imply that sampling is only involved after generating molecule from the shape?
* Efficiency is neither compared nor discussed. The authors make a fair argument in the introduction about the drawback of GEKO, which performs equally powerful as the proposed method. Given the proposed method also takes tons of training time, it would be good to see the comparison and discussion.
* Many discussions or relations to the literature are not discussed. e.g. how tokenization and lineraization are related to the literature of graph generation, framgment/scaffold-based molecule generation, etc.
* Experiments are only done over 12 protein targets, I would suggest adding more experiments, e.g. 100 targets in 3D SBDD paper.
* Ablation study is not complete, the comparison method, especially deep learning-based SBDD, 3D SBDD and liGAN are based on atomic reconstructions, it would be interesting to see whether the proposed model benefit from fragment-based method or the new training framework.
* Many details are missing or less mentioned,  For example, how the post-processing is done. How many bins are cut for the rotation and translation operations, where the origin is, how the tranformation is done, etc. In training & decoding section 2.4, only ligands data are mentioned while no protein data are mentioned. Figure 10 is not clear, not sure what shape, rotation, translation and category mean.
* Codes are not uploaded, which hinders the reproducibility of this work. It would be beneficial to the community if authors consider publishing the codes if the paper is accepted.

Overall, I think this paper makes a fair point and it is a good attempt in the direction of pocket-conditioned ligand design. I am willing to raise my score if the questions get answered.

---

> ### Author Response · Authors · 2022-08-02
> **Response to Reviewer Qdyd (part 1)**
>
> We thank reviewer Qdyd for giving constructive comments on our work. In the following paragraphs, we will answer the questions regarding the details of our model, the experiments, and the related work. Hope the replies can make our paper more clear. Further comments are welcome!
>
> **Method**
>
> **Q: How is sampling achieved for generating diverse molecules for a specific pocket? Is sampling only involved after generating molecules from the shape?**
>
> - The sampling is achieved in two steps: a) Sampling molecular shapes based on the given pocket. When sampling molecular shapes, we use different seed shapes and set the initial position of the seed shape randomly. Both of them contribute to the diversity of the generated molecules; 2) For each molecular shape, we further sample diverse molecules that fit it. Specifically, we employ the Nucleus decoding method to selectively combine different fragments in different decoding steps to achieve diversity. The sampling happens through the whole generation process. As reported in Table 1, as expected, our method obtained high diversity.
>
> **Q: How is the post-processing done?**
>
> - As mentioned in line 164, following our main competitor and previous state-of-the-art GEKO, the post-processing contains two steps: a) We remove the duplicate molecules. Specifically, if two generated molecules have the same SMILES, we randomly drop one of them; b) We further re-rank the generated molecules and eliminate the molecules that do not pass the affinity threshold.
>
> **Q: How many bins are cut for the rotation and translation operations? Where is the origin? How is the transformation done?**
>
> - Sorry for missing these details. We have added them to Appendix section 1.2. Thanks for pointing this out.
>
> - For rotation, the total number of bins is 8,712. To be precise, we enumerate 363 rotation axes in 3D space. For each axis, we enumerate 24 rotation angles. For the translation, the total number of bins is 21,952. In Appendix section 2.2, we have conducted several analytical experiments to study the discretization of these two operations. The results show that a) without discretization, the model can not generate molecules that fit the input shape, because of the non-linear relationship between quaternions and rotation angles b) with discretization, different bin sizes (7.5/15/30) do not have a significant difference.
>
> - While due to the trade-off between the granularity of the bin and the accuracy of the model, the number of the bin does not significantly affect the results.
>
> - For a fragment, we set its centroid as the origin. Because when handling a fragment, in order to align the same fragment in different 3D poses, we need to build up an internal coordinate that is not influenced by the external transformation. As we can determine the centroid of a fragment no matter what 3D pose it is, we treat it as the origin of the internal coordinate.
>
> - For rotation operation and translation operation, the transformation is done as follows:
>   - We represent the $i$-th rotation bin as a quaternion $q^{\mathrm{bin}}_i\in\mathbb{R}^{4}$. The discreterization of any continuous rotation operator $q \in\mathbb{R}^{4}$ can be computed by $\underset{i}{\arg \min}\|q^{\mathrm{bin}}_i, q\|_2$.
>   - We represent the $i$-th translation bin as the coordinate of its centre $t^{\mathrm{bin}}_i \in \mathbb{R}^{3}$. The discreterization of any continuous translation operator $t \in\mathbb{R}^{3}$ can be computed by $\underset{i}{\arg \min}\|t^{\mathrm{bin}}_i, t\|_2$.
>
> **Q: In training & decoding section 2.4, only ligands data is mentioned while no protein data is mentioned.**
>
> - In section 2.4, we do not mention the protein data. Because after sketching molecular shapes from the given protein, geometric information can be fully provided by the sketched shapes. In other words, we do not need other protein information for decoding molecules when the shape is given.
>
> **Q: What are shape, rotation, translation and category mean in Figure 10?**
>
> - Sorry for the confusion. We have fixed this in the new version of our draft.
>
> - *Shape* stands for the Shape Tanimoto [2], which measures the shape similarity between the input shape and generated molecules. *Rotation* stands for the accuracy of the model in predicting the correct rotation bin. *Translation* stands for the accuracy of the model in predicting the correct translation bin. *Category* stands for the accuracy of the model in selecting the correct fragment. All of them can be treated as metrics reflecting how well the model fits the data, which shows that our model builds up a strong mapping from shapes to molecules.

---

> ### Author Response · Authors · 2022-08-02
> **Response to Reviewer Qdyd (part 2)**
>
> **Experiment**
>
> **Q: Comparison of efficiency between proposed method and GEKO.**
>
> - As shown in Figure 7, DESERT is 20 times faster than GEKO in testing. Specifically, the figure demonstrates that GKEO has the slowest generation speed. DESERT-POCKET achieves similar performance but with a much faster speed. The reason is that GEKO employs a time-consuming docking process to train their MCMC-based model in a trial-and-error way. Instead, DESERT makes a more clever choice by pruning the space with its biological knowledge regarding the shape. Although DESERT needs two weeks for pretraining, the whole pretraining process is one-passed. For a novel pocket, we just use the previously trained models. Meanwhile, anytime we apply GEKO to a novel pocket, we have to train the model from scratch, which usually takes days.
>
> **Q: Apply the proposed method to the test data from 3D SBDD.**
>
> - We have reported the results in the general response, which demonstrates that our method can produce drug-like molecules with higher binding affinity. However, as we obtain the seed shape by overlapping multiple molecular shapes, the generated molecules tend to be structurally complicated, which leads to a moderately lower synthesis score.
>
> **Q: More ablation studies for analyzing the contribution of fragment-based style and pretraining framework.**
>
> - For the contribution of pretraining, as shown in Figure 10, when we increase the amount of pretraining data, our model achieves better performance on converting molecular shapes to 3D molecules, which implies that the pretraining component really helps the performance of our model. Although we also observe the model performance stops increasing when the size of dataset is over 4.5M, which makes sense because it is bounded by the model capacity and the problem complexity.
>
> - For the contribution of fragment-based generation style, we are training an atom-based DESERT for comparison. However, since the pretraining process is quite  time-consuming, we promise to  report it  and add discussions after obtaining the results.
>
> **Related Work**
>
> **Q: How tokenization and lineraization are related to the literature of graph generation, fragment/scaffold-based molecule generation, etc.**
>
> - Thanks. We have added related discussions to the appendix.
>   - We include discussions here about how the tokenization and linearization procedures relate to fragment-based drug design [3] [4].
>   - ***Fragment-based Drug Design*** Briefly, there are two approaches in FBDD: growing the fragment synthetically to a proximal binding site or linking two fragments together [4]. Our method can be classified as the previous type since the linearization generates a molecule in a one-by-one fashion.
>   - ***Tokenization*** The procedure is carefully designed for deep generative models to avoid loops and preserve functionalities, which is in spirit to the principle of [5]. Some FBDD work, such as [6] works in a discriminative approach. Thus there are not many constraints when cutting the molecules. Podda et al. [7] use the SMILES-fragment rather than a real molecule-fragment. Thus it can not utilize rich structured features.
>   - ***Linearization*** The procedure aims at traversing (or generating) a structured object in a left-to-right approach [8], which is tractable and scalable. It is borrowed from the area of computational linguistics, more specifically, syntactic parsing and structured generation [9] [10]. For micro-molecules, linearization (such as SMILES [11]) has been adopted for several decades. There is a line of research work for SMILES-based generation [12] [13] [14]. Similarly, in the area of macro-molecule, Huang et al. [15] designed a linear structure to estimate the likelihood of the structure of RNA.
>   - ***Comparison*** Compared with traditional linearized sequences such as SMILES [11], our method utilizes structural information to segment the molecules to preserve their functionality. Compared with topological generation based on a graph [5] [16] [17], our method is more scalable to big data since generating the variable graph topology is not friendly to large-batch training in neural networks.
>
> **Other**
>
> **Q: Codes are not uploaded, which hinders the reproducibility of this work.**
>
> - We have uploaded the core code of our method. We share the pre-trained checkpoint through an anonymous account for double-blindness: https://drive.google.com/file/d/1YCRORU5aMJEMO8hDT_o9uKCXmXTL5_5N/view?usp=sharing

---

> ### Author Response · Authors · 2022-08-02
> **Response to Reviewer Qdyd (part 3)**
>
> [1] Ari Holtzman et al., The Curious Case of Neural Text Degeneration, ICLR 2020
>
> [2] David Ryan Koes and Carlos J Camacho, Shape-based virtual screening with volumetric aligned molecular shapes, Journal of Computational Chemistry, 2014
>
> [3] Philine Kirsch et al., Concepts and Core Principles of Fragment-based Drug Design, Molecules, 2019
>
> [4] Christopher W Murray et al., The Rise of Fragment-based Drug Discovery, Nature Chemistry, 2009
>
> [5] Wengong Jin et al., Junction Tree Variational Autoencoder for Molecular Graph Generation, ICML 2018
>
> [6] Harrison Green et al., DeepFrag: A Deep Convolutional Neural Network for Fragment-based Lead Optimization, Chemical Science
>
> [7] Marco Podda et al., A Deep Generative Model for Fragment-based Molecule Generation, AISTATS 2020
>
> [8] Yijia Liu et al., Transition-based Syntactic Linearization, NAACL 2015
>
> [9] Yoon Kim et al., Unsupervised Recurrent Neural Network Grammars, NAACL 2019
>
> [10] Oriol Vinyals et al., Grammar as a Foreign Language, NeurIPS 2015
>
> [11] David Weininger, SMILES, A Chemical Language and Information System. 1. Introduction to Methodology and Encoding Rules, Journal of Chemical Information and Computer Sciences 1988
>
> [12] Matt J Kusner et al., Grammar Variational Autoencode, ICML 2017
>
> [13] Hanjun Dai et al., Syntax-directed Variational Autoencoder for Structured Data, ICLR 2018
>
> [14] Seokho Kang et al., Conditional Molecular Design with Deep Generative Model, Journal of Chemical Information and Modeling 2018
>
> [15] Liang Huang et al., LinearFold: Linear-time Approximate RNA Folding by 5’-to-3’Dynamic Programming and Beam Search, Bioinformatics 2019
>
> [16] Binghong Chen et al., Molecule Optimization by Explainable Evolution, ICLR 2020
>
> [17] Yutong Xie et al., Mars: Markov Molecular Sampling for Multi-objective Drug Discovery, ICLR 2021

---

> > ### Comment · Reviewer_Qdyd · 2022-08-03
> > **Thanks for the detailed answers**
> >
> > I appreciate the effort the authors made to clarify the points and improve the manuscript. Overall, I am happy with the response. Most of my concerns have been resolved in a good way. I have two further questions:
> >
> > * Does postprocessing happens before or after evaluation, it seems the postprocessing you explained may affect the evaluation result?
> > * The discretization of rotation quaternion and translation vector does not seem very intuitive to me. Could you elaborate more? It would also be interesting to see some discussions with other similar methods (e.g. Algorithm 23 in AlphaFold2 [1] Supplementary Material). Could this be a potential way to improve the performance further? It is okay at this point to leave as future work but it is interesting to see the discussion related to the literature.
> >
> > [1] Jumper, John, et al. "Highly accurate protein structure prediction with AlphaFold." Nature 596.7873 (2021): 583-589.

---

> > > ### Author Response · Authors · 2022-08-05
> > > **Further Response to Reviewer Qdyd (part 1)**
> > >
> > > Thanks a lot for your attention and the quick reply. We respond to the further two questions as follows:
> > >
> > > **Q: Does postprocessing happens before or after evaluation, it seems the postprocessing you explained may affect the evaluation result?**
> > >
> > > **A:**
> > >
> > > 1. Yes, the postprocessing happens before evaluation, which does affect the evaluation result. We include the postprocessing following GEKO (previous SOTA). We conduct experiments on GEKO's benchmark and employ the same postprocessing as GEKO for comparison.
> > > 2. We did a quick run on SBDD’s benchmark without post-processing (mentioned in your previous question) and find that without postprocessing (do not removing duplicate molecules and randomly selecting 100 molecules from DESERT's outputs for evaluation), the proposed DESERT still outperforms SBDD on 3 of 4 metrics. Note that DESERT works in a zero-shot way instead of using protein-ligand labeled data for training (the case of SBDD). Following are the detailed comparisons:
> > >
> > > - DESERT (w/o post-processing) achieved comparable (slightly better) Vina scores with 3D SBDD, SBDD employ pocket-ligand labeled data for training.
> > > - DESERT outperforms 3D SBDD on QED/Diversity.
> > > -  DESERT gives a lower SA score than 3D SBDD. As explained in the previous response to all reviewers, we assume that it is because the generated molecules of DESERT tend to be structurally complicated, which leads to a slightly worse synthesis score.
> > >
> > > In a word:
> > >   - In 3D SBDD's setting, DESERT generates slightly better results, **without any supervised data**.
> > >   - In GEKO's setting, DESERT generates SOTA results, **without any guidance during generation, but 20 times faster**.
> > >
> > >
> > >
> > > | Metric          | 3D SBDD              | DESERT-POCKET (w/o post-processing) |
> > > | --------------- | -------------------- | -------------------------- |
> > > | Vina (kcal/mol) | -6.069               | -6.148                     |
> > > | QED             | 0.522                | 0.614                      |
> > > | SA              | 0.672                | 0.612                      |
> > > | Diversity       | 0.873                | 0.926                      |
> > >
> > >
> > > Thanks again for your kind notes and we will add more discussions and comparisons to make this clear in our manuscript.
> > >
> > > **Q: The discretization of rotation quaternion and translation vector does not seem very intuitive to me. Could you elaborate more?**
> > >
> > > **A:**
> > >
> > > Yes, we would like to elaborate the discretization more clearly with some intuitive examples.
> > >
> > > - In terms of the **translation** vector, we show a simplified example in the 1-dimension space. Supposing the translation vector ranges from 0 to 10, we divide it into 5 bins: $[0, 2), [2, 4), [4, 6), [6, 8) $and $[8, 10]$. Given a translation vector 4.5, "discretization" means we put it into the 3rd bin -- $[4, 6)$.
> > > - The **rotation** quaternion can be expressed as a rotation of an angle $\theta^\circ$ around an axis $(x, y, z)$. Therefore, we discretize the quaternion in two steps: a) Enumerating rotation axes. For example, we can enumerate 8 rotation axes from the origin, i.e., $(0, 0, 1), (0, 1, 0), (0, 1, 1), (1, 0, 0)$, etc; b) Enumerating rotation angles for each axis. For example, we can enumerate the angle of every $15^\circ $. Combining the two steps, we can divide the range of quaternions into bins, like $(0, 0, 1, 0^\circ), (0, 0, 1, 15^\circ), \cdots, (0, 1, 1, 0^\circ), (0, 1, 1, 15^\circ)$, and so on. Given a quaternion $(0.1, 0.2, 0.9, 16^\circ)$, "discretization" means we map it to the 2nd bin -- $(0, 0, 1, 15^\circ)$.

---

> > > ### Author Response · Authors · 2022-08-05
> > > **Further Response to Reviewer Qdyd (part 2)**
> > >
> > > **Q: Further discussion about the difference with AlphaFold.**
> > >
> > > **A:**
> > >
> > > This is a very good question! We would like to firstly point out our opinions:**The main difference between our work and AlphaFold is whether to discretize the rotation quaternion**. Our key idea is to **avoid the discontinuity/ambiguity of quaternions when optimizing it** [5] [6].
> > >
> > > We would like to give a brief discussion here:
> > >
> > >   - **Overview**: There are several approaches to parameterize the rotation operator: quaternion [2], euler-angle [3], and SO(3) group (i.e., the rotation matrix) [4]. Quaternion and euler-angle are sometimes ambiguous and discontinous[5] [6] (see examples below).
> > >   - **Example of quaternion's ambiguity**: the rotation operator is periodic, rotating $180 \degree$ is equal to rotating $-180 \degree$, rotating $179.9 \degree$ is very close to rotating $-179.9 \degree$. Now considering a case when some object rotates $179.9 \degree$ and a neural network outputs $-179.9 \degree$, should it be penalized or not? Without discretization, the mean-square-error can be $[179.9 - (-179.9)]^2=359.8^2$ . However, with discretization, we convert a regression problem to a classification one, where $179.9 \degree$and $-179.9\degree$ is possible to lie in the same bin, i.e.,  $180\degree$.
> > >   - **How AlphaFold avoids such ambiguity**: In AlphaFold, the quaternion is an intermediate variable. AlphaFold **does not optimize the quaternion directly** (AlphaFold's Appendix 1.9.3), thus it avoids such an issue.  Directly optimizing the quaternion in a regression way may hurt the performance [5] [6], similar to our observation (Figure 3, Appendix).
> > >   - **Future work**: In the area of structural biology, some researchers prefer to optimize two rows of a rotation matrix, instead of the quaternion [7]. We will leave this for future work.
> > >
> > > [1] Jumper, John, et al. "Highly accurate protein structure prediction with AlphaFold." Nature 596.7873 (2021): 583-589.
> > >
> > > [2] *https://en.wikipedia.org/wiki/Quaternion*
> > >
> > > [3] *https://en.wikipedia.org/wiki/Euler_angles*
> > >
> > > [4] *https://en.wikipedia.org/wiki/SO3*
> > >
> > > [5] Zhou Y, Barnes C, Lu J, et al. On the continuity of rotation representations in neural networks[C]//Proceedings of the IEEE/CVF Conference on Computer Vision and Pattern Recognition. 2019: 5745-5753.
> > >
> > > [6] Falorsi L, De Haan P, Davidson T R, et al. Explorations in homeomorphic variational auto-encoding[J]. arXiv preprint arXiv:1807.04689, 2018.
> > >
> > > [7] Zhong E D, Bepler T, Berger B, et al. CryoDRGN: reconstruction of heterogeneous cryo-EM structures using neural networks[J]. Nature methods, 2021, 18(2): 176-185.

---

> > > > ### Comment · Reviewer_Qdyd · 2022-08-05
> > > > **Thanks for your clarification**
> > > >
> > > > Thanks for the detailed discussion. It may be worth trying to adopt AlphaFold's way to handle quaternion as well in future work which could avoid discretization. Overall, I am happy with your rebuttal. Thanks for the effort to improve the manuscript a lot. I am revising my score accordingly.

---

### Official Review · Reviewer_fWm8 · 2022-07-09

**Rating:** 4
**Confidence:** 4
**Soundness:** 2 fair
**Presentation:** 2 fair
**Contribution:** 2 fair

**Summary:**

This paper proposes a zero-shot drug design method based on sketch. Starting from the property that structure determines properties, the paper uses a sketch model of protein pocket shapes and a pre-trained generative model based on molecular shape sketches to achieve zero-shot drug design that does not rely on docking experimental data and docking simulations.

**Questions:**

1.	Is there a theoretical basis for using the intersection of a seed shape and a pocket shape to obtain a molecule shape? How are the shape, size, and initial position of the seed shape chosen in the algorithm? Different initial parameter settings will result in very different molecule shapes and thus completely different generated molecules. How do different initial parameters affect the generated results, and is this approach reasonable?
2.	The model is encoded using discretized spatial information, while the generation directly generates the 3D coordinates of the molecular functional groups. Does the model then still generate the same or at least similar molecules when given a molecule shape that has been rotated and translated? This property is necessary for the generation of physical entities.
3.	Although the pre-trained model uses a larger molecular dataset, not all molecules in the ZINC database are used for pharmaceutical purposes as far as I know. The molecules generated by the pre-training model may not be suitable for use as drugs. Measurements of drug-like properties(QED), ease of synthesis(SA) and lipid solubility indicators(LogP) and comparisons with other models should be included in the experimental section.
4.	It is mentioned in the paper that the pre-trained model uses a large amount of computational resources, therefore the source code and pre-trained model should be made public for easy reproduction.


**Limitations:**

Using only information about the shape of the protein pocket avoids some of the limitations of the data set and binding simulations. However, the model is limited by ignoring information about the different interaction forces (hydrogen bonding, π-Stacking, etc.) that occur when molecules bind to proteins depending on the type of atom since it completely abandoned such information.

During the generation phase, the tree-like structure is less expressive for molecules with complex structures. Moreover, the connecting step based entirely on greedy algorithms may lead to unreasonable results.


**Strengths And Weaknesses:**

Strengths:
The paper presents a molecular generation method based only on shape sketching, which is a very novel approach. Also, the generation method without using protein binding data and simulations is enlightening for future work.

Weakness:
	Although the sketch-based generation idea is novel, the generation model is still similar to the transformer-based sequence generation model used in machine translation tasks. The paper is also not clear enough in the method description, only the design of the encoding and decoding method and the paper citation of the model used are given, the specific model architecture as well as the parameters should be given.

---

> ### Author Response · Authors · 2022-08-02
> **Response to Reviewer fWm8 (part 1)**
>
> Thanks for your valuable comments! We will answer your questions regarding the proposed method, and the experiment setting respectively in the following paragraphs. Further comments are welcome!
>
> **Method**
>
> **Q: The generation model is still similar to the transformer-based sequence generation model used in machine translation tasks, and specific model architecture, as well as the parameters, should be given.**
>
> - Yes, we use the widely used transformer architecture for sequence generation [1] [2]. However, which is not the main contribution of our paper. The core contribution of our work is using massive unbound molecules to pretrain the drug design model in a zero-shot fashion, which is model-agnostic. In practice, the architecture is free to change.
>
> - From line 154 to 158, we have listed many details about our model, including the number of network layers, model dimension, batch size, learning rate, etc. We also add other parameter details, including the number of attention heads, and patch size, in Appendix section 1.3 of the new version. Thanks for pointing out this.
>
> **Q: Is there a theoretical basis for using the intersection of a seed shape and a pocket shape to obtain a molecule shape?**
>
> - As we mentioned in section 1, DESERT is not baseless. We design the intersection strategy based on two principles: a) Structure determines properties. [3] [4] [5] show a drug candidate would have satisfactory bio-activity to a target pocket if their shapes are complementary. b) Ligand often attaches tight to a pocket. As we mentioned in line 69 and Figure 1, we have conducted several preliminary studies, which show the average distance between ligands and pockets is $1.52A$, even less than the length of C-C bond, i.e., $1.54A$, in a molecule itself. Based on these principles, our desired molecular shapes should satisfy the property, i.e., complementary to the pocket, to achieve good bioactivity. The intersection method makes the sketched molecular shape meet the requirement.
>
> - The intersection method makes the sketched molecular shape meet the requirement due to two premises: a) Two shapes complement each other if part of their boundaries matches. b) The intersection method ensures that the generated shape shares some boundary with the pocket's shape.
>
> **Q: How are the shape, size, and initial position of the seed shape chosen in the algorithm?**
> - Thanks for pointing out these missing details. We have added them to Appendix section 1.1.
>
> - We get the seed shape by heuristically overlapping the shapes of several drug-like molecules sampled from ZINC. Our desired molecular shapes should satisfy two properties: a) Complement to the pocket to achieve good bioactivity, which means part of their boundaries are close to each other; b) Be a drug-like shape (e.g., not a rectangular solid) and not overly dependent on one specific molecule for diversity. The property a) is satisfied since the boundary of the intersected area matches some part of the pocket's boundary. The property b) is satisfied by overlapping molecules' shapes to avoid generating odd shapes, such as rectangle or triangle shapes which never occur in molecules. The results show that the overlapping method is relatively effective.
>
> - Because we obtain the seed shape by overlapping drug-like molecules, the size of the seed shape is decided by the sampled molecules.
>
> - For the initial position, we randomly sample one as long as the seed shape is outside the pocket shape. With such strategies, we can explore different regions of a given pocket, making our method produce diverse molecules.
>
> **Q: How do different initial parameters of the seed shape affect the generated results?**
>
> - In Appendix section 2.2, we discuss the influence of different types of seed shapes on the model performance. Compared with using the entire pocket directly, using a seed shape achieves a better binding affinity. The results indicate that the seed shape can capture protein's structural information more moderately.
>
> - In section 3.5, we also discuss how the number of molecular shapes sampled with the seed shape affects the method's performance. In Figure 11, we find that increasing the number gives us a performance rise, which implies comprehensive explorations of pockets benefits model performance.
>
> **Q: Does the model then still generate the same or at least similar molecules when given a molecule shape that has been rotated and translated?**
>
> - Following liGAN, when training the model, we randomly rotate and translate the molecules to make our model have rotation and translation invariance ability. We compare the similarity of generated molecules based on the different or same molecular shapes (both randomly rotate and translate). The similarity rises from 0.092 to 0.508, which shows that with the same molecular shape as input, the model produces similar molecules as expected. Sorry for missing the details. We have added them to our manuscript.

---

> ### Author Response · Authors · 2022-08-02
> **Response to Reviewer fWm8 (part 2)**
>
> **Q: Not all molecules in the ZINC database are used for pharmaceutical purposes.**
>
> - Thanks for pointing this out. We only use the drug-like subset of ZINC to train our model. We have made this clear in our manuscript.
>
> **Q: The tree-like structure is less expressive for molecules with complex structures.**
>
> - The tree-like structure is expressive enough. We did a quick run and found that tree-like structures can describe over 96% of molecules in the ZINC database. Specifically, we sample 10M molecules from ZINC drug-like subset and analyze their structure after fragment cutting. We find that 62% of these molecules have a native sequence structure, and 34% of molecules only have one branch. Based on these results, we think that the tree-like structure is expressive enough.
>
> **Q: The model is limited by ignoring information about the different interaction forces.**
>
> - As mentioned in Appendix section 2.2, our method supports considering information about the interaction forces in the decoding phase of our method and we even conduct some preliminary experiments. However, the preliminary results do not give positive results. The reason might be that interaction forces used in our preliminary experiments do not fit our shape-based pretrained model and maybe we can include more chemical information into consideration, e.g., the bond length (which can be our future work). However, this shows that our model is not limited by ignoring interaction force information.  Nevertheless, only using geometric information makes our method outperform previous work, and we leave that utilizing interaction force information as future work.
>
> **Experiment**
>
> **Q: Measurements of drug-like properties(QED), ease of synthesis(SA) and lipid solubility indicators(LogP) and comparisons with other models should be included in the experimental section.**
>
> - As mentioned in section 3.1 and Appendix section 2.1, following [6] [7] [8], we combine QED and SA to build the metric Succ. Specifically, the Succ tells us the percentage of generated molecules that satisfy a widely used rule of thumb [9], i.e., QED >= 0.25, SA >= 0.59, and Vina score <= -8.18. Moreover, we also include these three metrics individually in our new experiment, whose results are shown in our general response.
>
> **Other**
>
> **Q: The source code and pre-trained model should be made public for easy reproduction.**
>
> - We have uploaded the core code of our method. We share the pre-trained checkpoint through an anonymous account for double-blindness: https://drive.google.com/file/d/1YCRORU5aMJEMO8hDT_o9uKCXmXTL5_5N/view?usp=sharing
>
> [1] Chloe Hus et al., Learning Inverse Folding from Millions of Predicted Structures, ICML 2022
>
> [2] Roshan Rao et al., MSA Transformer, ICML 2021
>
> [3] Alan R Katritzky et al., Computational Chemistry Approaches for Understanding How Structure Determines Properties, Zeitschrift für Naturforschung B, 2009
>
> [4] Geza Gruenwald, Plastics: How Structure Determines Properties, 1992
>
> [5] Alan R. Katritzky et al., How Chemical Structure Determines Physical, Chemical, and Technological Properties:  An Overview Illustrating the Potential of Quantitative Structure−Property Relationships for Fuels Science, Energy Fuels, 2005
>
> [6] Yuwei Yang et al., Knowledge Guided Geometric Editing for Unsupervised Drug Design. 2022
>
> [7] Yutong Xie et al., MARS: Markov Molecular Sampling for Multi-objective Drug Discovery, ICLR 2021
>
> [8] Wengong Jin et al., Multi-Objective Molecule Generation using Interpretable Substructures, ICML 2020
>
> [9] Oleg Ursu et al., DrugCentral 2018: an update, Nucleic acids research, 2019

---

> ### Author Response · Authors · 2022-08-06
> **Response to Reviewer fWm8 (part 3)**
>
> Dear Reviewer, we appreciate your valuable advice, which helps improve our manuscript a lot.
>
> Did our response and the updated manuscript address your questions? We are happy to discuss any further concerns.
>
> Thanks for your time!

---

> ### Author Response · Authors · 2022-08-09
> **Response to Reviewer fWm8 (part 4)**
>
> Thanks again for your comments. Did we fix your concern of this paper properly? If not, we are happy to take further questions!

---

### Official Review · Reviewer_M9mc · 2022-07-11

**Rating:** 6
**Confidence:** 4
**Soundness:** 3 good
**Presentation:** 2 fair
**Contribution:** 3 good

**Summary:**

This paper proposes a 3D molecular generative model DESERT for structure-based drug design in a zero-shot manner. The method involves two stages: first sketching the molecular shape in the protein pocket, then generating molecules based on the shape. Based on the assumption of "structure determines properties", the method aims to find molecules whose shapes are complementary to the pocket and thus massive unbounded molecular data (e.g. ZINC)  can be utilized to train the shape2mol model.

**Questions:**

- I'm a bit confused about the sketching part (Sec. 2.2). How do you get the seed shape? In algorithm 1 in appendix, how is the volume threshold t and step size alpha determined?

- In Sec. 2.3, what is the output of shape encoder and the input of shape decoder? How is the spatial correspondency established in the proposed network architecture?  Is there any guarantee that the generated molecule will satisfy the shape constraint?
- Which resolution is the shape voxelized at (Sec. 2.3.1)?  Since the number of voxels increases cubically to the pocket size, will it have the scalability issue?
- In line 167, the authors stated "For each protein pocket, we sketch 200 shapes. For each shape, we generate 1000 molecules". In line 191, it seems all other baselines methods "generate 100 molecules for comparison".  I'm wondering how the samples generated by the proposed method (200 x 1000) are utilized to compute scores in Table 1 for a fair comparison with other existing methods? Also, "20 times faster than GEKO" refers to a per-pocket or per-sample inference time?
- Although the proposed method is based on the assumption that "structure determines properties", I'm still curious about how the generated molecules could have so good Vina scores without any protein pocket information leveraged in the generation process.

**Limitations:**

The authors discuss some limitations in the experiment section. There is not negative social impact

**Strengths And Weaknesses:**

Strengths:
- The main idea of drug design by sketching and generating is novel and well-motivated
- The proposed method shows better performance over existing methods in the experiments
- The analysis in experiments is comprehensive


Weaknesses:
- Don't have a related work section discussing the connection between proposed Shape2Mol and existing shape-based molecular generation methods (Ref. [49] - [53])
- The proposed method section is not very clear to me. See details in Questions

---

> ### Author Response · Authors · 2022-08-02
> **Response to Reviewer M9mc (part 1)**
>
> We thank reviewer M9mc for the helpful suggestions. Following the suggestions, we clarify the details of our method. We also address these issues below. Further comments are welcome!
>
> **Method**
>
> **Q: How do you get the seed shape? How is the volume threshold t and step size alpha determined?**
>
> - We are sorry for missing these details. We have added them to Appendix section 1.1. Thanks for pointing this out.
>
> - We get the seed shape by heuristically overlapping the shapes of several drug-like molecules sampled from ZINC. Our desired molecular shapes should satisfy two properties: a) Complement the pocket to achieve good bioactivity, which means part of their boundaries are close to each other; b) Be a drug-like shape (e.g., not a rectangular solid) and not overly dependent on one specific molecule for diversity. The property a) is satisfied since the boundary of the intersected area matches some part of the pocket's boundary. The property b) is satisfied by overlapping molecules' shapes to avoid generating odd shapes, such as rectangle or triangle shapes which never occur in molecules. The results show that the overlapping method is relatively effective.
>
> - For the volume threshold, we compute the averaged volume of some molecules, i.e., $300A^3$. The number can be viewed as an estimator of the expectation of a molecule's size. Through this, we can avoid generating too large or too small shapes.
>
> - We set the step size as $0.5A$ because it can be reflected by the voxelized shapes, whose resolution is $0.5A$ too.
>
> **Q: What is the output of shape encoder and the input of shape decoder?**
>
> - Thanks for pointing this out. We have added both of them to Appendix 1.3.
>
> - The output of the shape encoder is the continuous representation of each 3D patch, which contains the geometric information of inputted molecular shape. It will serve as the context of the decoder to constrain the shape of generated molecules.
>
> - The input of the shape decoder in decoding step *t* is the fragment category, rotation quaternion, and translation vector from the decoder output at time *t-1*. We use them to tell the model how exactly a fragment is placed in 3D space so that the model can generate the next fragment connected with it. The output of the shape encoder is also inputted as the geometric context of decoding.
>
> **Q: How is the spatial correspondency established in the proposed network architecture? Is there any guarantee that the generated molecule will satisfy the shape constraint?**
>
> - We established correspondence by the powerful neural networks trained on large-scale data. Note that there is no guarantee theoretically. However, as we mentioned in section 3.3, the good results of Shape Tanimoto [2] suggest that generated molecules satisfy the shape constraint empirically.
>
> **Q: Which resolution is the shape voxelized at, and will it cause the scalability issue when the pocket size increases?**
>
> - The resolution of the voxelized shape is 0.5A. As the length of the most common chemical bond, i.e., the C-C bond, is $1.54A$, the resolution is clear enough to describe the molecular shape.
>
> - We avoid the scalability issue by using two techniques: a) Limit the maximum number of voxels with a spanned cube. Following liGAN, we only apply the voxelization in the cube around the given pocket, which detaches the connection between the voxel number and pocket size; b) As we mentioned in 2.3.1, we further use the 3D patch to compress the number of voxels. By using the technique, a group of voxels is compressed and processed together, which can let us handle a large number of voxels. In particular, we compress 21,952 voxels to 343 patches in our paper.
>
> **Experiment**
>
> **Q: How do you reduce the number of molecules in the experiments?**
>
> - As mentioned in section 2.4, we reduce the number of molecules in two steps: a) Re-rank the molecules. Following our main competitor GEKO, we use vina local energy minimization to re-rank the generated molecules; b) Drop the unwanted molecules. After the re-ranking, we only keep the top 100 molecules in our experiments.
>
> **Q: "20 times faster than GEKO" refers to a per-pocket or per-sample inference time?**
>
> - It refers to the per-pocket case. To calculate the speed, we measure the time which starts from the given pocket to the end of getting the 100 molecules. We have made it clear in our manuscript.
>
> **Q: How the generated molecules could have so good Vina scores without any protein pocket information leveraged in the generation process?**
>
> - Actually, as shown in Figure 3, the pocket information is used in the generation process. When we design molecules based on a given pocket, we sample the molecular shape from the pocket, which contains the geometric information of the pocket. As we reported in section 3.2, the shape helps DESERT produce high-quality molecules.

---

> ### Author Response · Authors · 2022-08-02
> **Response to Reviewer M9mc (part 2)**
>
> **Related Work**
>
> **Q: Related work section discussing the connection between proposed Shape2Mol and existing shape-based molecular methods.**
>
> - Here we discuss the relationship between our method and existing shape-based drug design [3] approaches. Some previous work designed new drugs based on the shape of a known ligand. Traditional approaches work in a retrieval way, i.e., finding molecules whose shape is most similar to a known one [4] [5]. Modern deep learning models can decode a molecule from its shape [6] [1]. Such ligand-based generation can not generalize to unseen pockets. [7] directly generates molecules from the pocket shape, which is the most closed to our work. However, our model works in a fragment-based fashion, while theirs works in an atomic way. What makes our model especially different is that we utilize the power of the pre-training model to make pocket-based drug design more promising.
>
> [1] Miha Skalc et al., Shape-Based Generative Modeling for de Novo Drug Design, JCIM 2019
>
> [2] Koes et al., Shape-based Virtual Screening with Volumetric Aligned Molecular Shapes, Journal of computational chemistry, 2014
>
> [3] Montfort et al., Structure-based Drug Design: Aiming for a Perfect Fit, Essays in biochemistry, 2017
>
> [4] Kumar et al., Advances in the development of shape similarity, Frontiers in chemistry, 2018
>
> [5] Santos et al., Drug Screening using Shape-based Virtual Screening and in vitro Experimental Models of Cutaneous Leishmaniasis, Parasitology, 2021
>
> [6] Masuda et al., Generating 3D Molecular Structures Conditional on a Receptor Binding Site with Deep Generative Models, NeurIPS 2020
>
> [7] Shitong Luo et al., A 3D Generative Model for Structure-Based Drug Design, NeurIPS 2021

---

> > ### Comment · Reviewer_M9mc · 2022-08-07
> > **Thank you for the response**
> >
> > Thank you for the detailed responses and revisions. I still have some questions about the model and your supplementary experiments compared to 3D 3BDD.
> > * Does the decoded output at each step correspond to a specific 3D patch (the input of the encoder)? If the decoding is unordered, how does it align with your tree linearization algorithm (is there any guarantee that the decoded fragment sequence is a valid tree)?
> > * If I understand it correctly, during the sampling phase, you sketch 200 shapes and generate 1000 molecules for each shape (which means 200k sampled molecules), then rank them using vina local energy minimization and select the top 100 molecules. Is it same in supplementary experiments compared to 3D SBDD?  LiGAN and 3D SBDD only randomly sampled 100 molecules, such a comparison would be unfair.  Also, since Vina score is one of the evaluation metrics, I don't think it should be used as the ranking criterion. (It may be fine when compared to GEKO, since it uses Vina score as the training signal.)

---

> > > ### Author Response · Authors · 2022-08-07
> > > **Further Response to Reviewer M9mc (part 1)**
> > >
> > > Thanks a lot for your further comments on our response. Hope our following answers can fix your concerns. If not, any further questions are welcome!
> > >
> > > **Q: Does the decoded output at each step correspond to a specific 3D patch (the input of the encoder)?**
> > >
> > > **A:**
> > >
> > > - No, although the decoded output at each step is also a 3D object (3D molecular fragment), which could not explicitly correspond to a specific 3D patch of the encoder input.
> > > - Note that our proposed DESERT works totally in an end-to-end way, which does not include any obligatory and explicit correspondence between the encoding input (3D patch) and the decoding output (3D molecular fragment).
> > > - We think such end2end learning is very intuitive and widely exists, which learns the implicit correspondence via neural networks and large scaled data. For example, in the standard end2end (sequence-to-sequence) learning task machine translation (e.g., English to German), the output German word at each step does not have explicit correspondence (called alignment in translation) to the input English words, but it gives good empirical translation results. The correspondence could be predicted according to the intermediate attention parameters.
> > > - The proposed DESERT gives good correspondence results empirically. As we reported in section 3.3, the Shape Tanimoto between input shapes and generated molecules is 0.875 (maximum value is 1.0), which indicates that DESERT can well learn the correspondence between the generated molecules and input shapes.
> > >
> > >
> > > **Q: If the decoding is unordered, how does it align with your tree linearization algorithm (is there any guarantee that the decoded fragment sequence is a valid tree)?**
> > >
> > > **A:**
> > >
> > > - Do you mean our decoder works in a non-autoregressive way? No, The decoding is ordered. Similar to [1] [2] [3] [4] [5], we generate the fragment sequence in left-to-right order.
> > > - Although there is no theoretical guarantee for generating a valid tree, we find the proposed DESERT model rarely generates invalid outputs empirically.
> > > For example, 95.0% of generated sequences can be converted to valid molecules in our experiments on SBDD's test data. (95% is the percentage of generated molecules that can pass the validity check of RDKit)
> > > - Practically, we just drop the invalid outputs for convinience. Most of the invalid cases are caused by the valence error, i.e., the number of chemical bonds attached to an atom is larger than the atom can have. The error can be moderated by imposing constraints on the number of new branches at a splitting node.
> > >
> > > **Q: Is it same in supplementary experiments compared to 3D SBDD? LiGAN and 3D SBDD only randomly sampled 100 molecules, such a comparison would be unfair. Also, since Vina score is one of the evaluation metrics, I don't think it should be used as the ranking criterion. (It may be fine when compared to GEKO, since it uses Vina score as the training signal.)**
> > >
> > > **A:**
> > >
> > > Thanks for your kind notes. Very good question, which has also been mentioned by Reviewer Qdyd. We conduct experiments on GEKO's benchmark and follow the same postprocessing (using Vina for reranking) as GEKO for comparison. We totally agree that it is not appropriate to compare the proposed DESERT and 3D SBDD in such a setting in the supplementary experiments.
> > >
> > > To fix the concern, we did a quick run on SBDD's benchmark and find that **DESERT outperforms 3D SBDD without the reranking process**.
> > >
> > > We conduct experiments under two settings to make comparisons between 3D SBDD and DESERT:
> > >
> > > 1.  We **remove the post-processing step of DESERT**, and compare it with SBDD.
> > > 2.  We **add the same post-processing step to SBDD** by drawing the same number of molecules (200k) as DESERT. Similar to DESERT, we use the released code of SBDD and set `num_samples=200000`, then use Vina to select the top-100 molecules for comparison.
> > >
> > > Results show that:
> > >
> > > | Metric| 3D SBDD (w/o post-processing)| 3D SBDD (w post-processing)             | DESERT-POCKET (w/o post-processing) | DESERT-POCKET (w post-processing) |
> > > | ------- | ----------|---------- |--------- |--------- |
> > > | Vina (kcal/mol) | -6.069      | -7.584    | -6.148  | -9.410  |
> > > | QED             | 0.522       | 0.501     | 0.614   | 0.549   |
> > > | SA              | 0.672       | 0.623     | 0.612   | 0.616   |
> > > | Diversity       | 0.873       | 0.826     | 0.926   | 0.908   |
> > >
> > >
> > > **DESERT outperforms 3D SBDD in both with/without post-processing setting on 3 of 4 metrics: Vina, QED and Diversity.** Note that DESERT works in a zero-shot way instead of using protein-ligand labeled data for training (the case of SBDD).
> > >
> > > DESERT gives a lower SA score than 3D SBDD. As explained in the previous response to all reviewers, we assume that it is because the generated molecules of DESERT tend to be structurally complicated, which leads to a slightly worse synthesis score.
> > >
> > > Thanks again for pointing out the concern in the experimental comparison. We will fix it throughout the whole paper to make it clear.

---

> > > > ### Comment · Reviewer_M9mc · 2022-08-07
> > > > **Thanks for the response**
> > > >
> > > > Thanks for the response. I understand such style of model and training approach is widely used in domains like machine translation, but I'm still a bit surprised that the spatial constraint could be implicitly handled in the neural network by simply predicting the discretized rotation quaternion and translation vector without mapping them back in the 3D space during the sampling phase. I think it would be an interesting future direction to include some important inductive bias like roto-translational equivariance in the modeling.
> > > >
> > > > In summary, my concerns are mostly addressed and I remain in favor of accepting this paper.

---

> > > ### Author Response · Authors · 2022-08-07
> > > **Further Response to Reviewer M9mc (part 2)**
> > >
> > > [1] David Weininger, SMILES, A Chemical Language and Information System. 1. Introduction to Methodology and Encoding Rules, Journal of Chemical Information and Computer Sciences 1988
> > >
> > > [2] Marco Podda et al., A Deep Generative Model for Fragment-based Molecule Generation, AISTATS 2020
> > >
> > > [3] Matt J Kusner et al., Grammar Variational Autoencode, ICML 2017
> > >
> > > [4] Hanjun Dai et al., Syntax-directed Variational Autoencoder for Structured Data, ICLR 2018
> > >
> > > [5] Seokho Kang et al., Conditional Molecular Design with Deep Generative Model, Journal of Chemical Information and Modeling 2018

---

> ### Author Response · Authors · 2022-08-06
> **Response to Reviewer M9mc (part 3)**
>
> Dear Reviewer, we appreciate your valuable advice, which helps enhance our manuscript.
>
> We are happy to discuss if you have any further suggestions or concerns.
>
> Thanks for your time!

---

### Author Response · Authors · 2022-08-02
**Response to all the reviewers and area chairs**

Thanks for your valuable comments! We are encouraged that all the reviewers consider our work novel, and we are also sorry for missing some details of the proposed method and experiments. We have updated our submission (with red text) to clarify our approach. For your convenience, we have also listed the changes here:

- In Appendix 2.2, to prove the effectiveness of DESERT more widely, we add experimental results on the test data from 3D SBDD [1], as mentioned by Reviewer Qdyd.

  - In general, DESERT-POCKET has better results on 4 of 5 benchmarks.
  DESERT-POCKET achieves a remarkably higher Vina score (-9.377 vs -6.344 [3D SBDD]) and higher affinity (87.30 vs 29.09 [3D SBDD]). The reasons are two folds: a) the pretraining process lets the model have the ability to generate plausible molecules given any shape, whereas 3D SBDD only learns to generate molecules from a limited number of pockets. b) our model adopts more diverse sampling techniques, while the diversity of 3D SBDD is constrained by inputting a unique pocket shape.
  The SA is a bit lower yet acceptable (SA>0.59 is considered good in GEKO). We assume it is because the generated molecules tend to be more structurally complicated.


  | Metric        || liGAN  | 3D SBDD | DESERT-POCKET * | Ref    |
    | ------------- | ------ | ------- | --------------- | ------ | ------ |
    | Vina          | Avg.   | -6.144  | -6.344          | -9.377 | -7.158 |
    || Med.          | -6.100 | -6.200  | -9.410          | -6.950 |        |
    | QED           | Avg.   | 0.371   | 0.525           | 0.554  | 0.484  |
    || Med.          | 0.369  | 0.519   | 0.549           | 0.469  |        |
    | SA            | Avg.   | 0.591   | 0.657           | 0.614  | 0.733  |
    || Med.          | 0.570  | 0.650   | 0.616           | 0.745  |        |
    | High Affinity | Avg.   | 23.77   | 29.09           | 87.30  | -      |
    || Med.          | 11.00  | 18.50   | 96.00           | -      |        |
    | Diversity     | Avg.   | 0.655   | 0.720           | 0.908  | -      |
    || Med.          | 0.676  | 0.736   | 0.908           | -      |        |
    | logP          | Avg.   | -       | -               | 3.163  | -      |
    || Med.          | -      | -       | 3.224           | -      |        |

  - As the test data from 3D SBDD [1] only provide incomplete protein pockets, we recover the complete pockets by aligning the incomplete protein to the structures from the PDB database. Finally, we managed to recover 55 pockets (a little difficult data pre-processing process) and apply DESERT on them. The results of liGAN and 3D SBDD are from [1]

- In Appendix section 3.1, we discuss the connection between our work and previous shape-based methods

- In Appendix section 3.2, we discuss the connection between our work and previous fragment-based methods.

- In Appendix section 1.3, we add a more detailed description of our model's hyperparameters.

[1] Luo et al., A 3D Generative Model for Structure-Based Drug Design, NeurIPS 2021.

---

### Meta-Review · Area_Chair_iP21 · 2022-08-26

**Recommendation:** Accept
**Confidence:** Certain

**Metareview:**

The paper makes a novel contribution to methods for generating novel molecules from scratch. The core idea is to generate a shape that fits the molecular pocket without looking at the protein structure.

Two out of three reviewers recommended acceptance. Reviewers emphasize that the method is innovative and interesting, and the empirical performance appealing (especially given that only the shape information is provided to the model). Strong performance is enabled by good design choices made across the paper, such as including the pretraining stage.

The reviewer that recommend rejection raised issues related to the novelty and clarity of the paper. However, I believe the paper is sufficiently clear and novel to meet the bar for acceptance.

Overall, it is my pleasure to recommend acceptance of the paper.

**Award:**

No

---

### Decision · Program_Chairs · 2022-09-14

Accept